# FASTTEXT.ZIP:
# COMPRESSING TEXT CLASSIFICATION MODELS

**Armand Joulin, Edouard Grave, Piotr Bojanowski, Matthijs Douze, Hervé Jégou & Tomas Mikolov**
Facebook AI Research
`{ajoulin,egrave,bojanowski,matthijs,rvj,tmikolov}@fb.com`

## ABSTRACT

We consider the problem of producing compact architectures for text classification, such that the full model fits in a limited amount of memory. After considering different solutions inspired by the hashing literature, we propose a method built upon product quantization to store word embeddings. While the original technique leads to a loss in accuracy, we adapt this method to circumvent quantization artefacts. Combined with simple approaches specifically adapted to text classification, our approach derived from `fastText` requires, at test time, only a fraction of the memory compared to the original FastText, without noticeably sacrificing quality in terms of classification accuracy. Our experiments carried out on several benchmarks show that our approach typically requires two orders of magnitude less memory than `fastText` while being only slightly inferior with respect to accuracy. As a result, it outperforms the state of the art by a good margin in terms of the compromise between memory usage and accuracy.

## 1 INTRODUCTION

Text classification is an important problem in Natural Language Processing (NLP). Real world use-cases include spam filtering or e-mail categorization. It is a core component in more complex systems such as search and ranking. Recently, deep learning techniques based on neural networks have achieved state of the art results in various NLP applications. One of the main successes of deep learning is due to the effectiveness of recurrent networks for language modeling and their application to speech recognition and machine translation (Mikolov, 2012). However, in other cases including several text classification problems, it has been shown that deep networks do not convincingly beat the prior state of the art techniques (Wang & Manning, 2012; Joulin et al., 2016).

In spite of being (typically) orders of magnitude slower to train than traditional techniques based on n-grams, neural networks are often regarded as a promising alternative due to compact model sizes, in particular for character based models. This is important for applications that need to run on systems with limited memory such as smartphones.

This paper specifically addresses the compromise between classification accuracy and the model size. We extend our previous work implemented in the `fastText` library[1]. It is based on n-gram features, dimensionality reduction, and a fast approximation of the softmax classifier (Joulin et al., 2016). We show that a few key ingredients, namely feature pruning, quantization, hashing, and re-training, allow us to produce text classification models with tiny size, often less than 100kB when trained on several popular datasets, without noticeably sacrificing accuracy or speed.

We plan to publish the code and scripts required to reproduce our results as an extension of the `fastText` library, thereby providing strong reproducible baselines for text classifiers that optimize the compromise between the model size and accuracy. We hope that this will help the engineering community to improve existing applications by using more efficient models.

This paper is organized as follows. Section 2 introduces related work, Section 3 describes our text classification model and explains how we drastically reduce the model size. Section 4 shows the effectiveness of our approach in experiments on multiple text classification benchmarks.

---

[1]https://github.com/facebookresearch/fastText

## 2 RELATED WORK

**Models for text classification.** Text classification is a problem that has its roots in many applications such as web search, information retrieval and document classification (Deerwester et al., 1990; Pang & Lee, 2008). Linear classifiers often obtain state-of-the-art performance while being scalable (Agarwal et al., 2014; Joachims, 1998; Joulin et al., 2016; McCallum & Nigam, 1998). They are particularly interesting when associated with the right features (Wang & Manning, 2012). They usually require storing embeddings for words and n-grams, which makes them memory inefficient.

**Compression of language models.** Our work is related to compression of statistical language models. Classical approaches include feature pruning based on entropy (Stolcke, 2000) and quantization. Pruning aims to keep only the most important n-grams in the model, leaving out those with probability lower than a specified threshold. Further, the individual n-grams can be compressed by quantizing the probability value, and by storing the n-gram itself more efficiently than as a sequence of characters. Various strategies have been developed, for example using tree structures or hash functions, and are discussed in (Talbot & Brants, 2008).

**Compression for similarity estimation and search.** There is a large body of literature on how to compress a set of vectors into compact codes, such that the comparison of two codes approximates a target similarity in the original space. The typical use-case of these methods considers an indexed dataset of compressed vectors, and a query for which we want to find the nearest neighbors in the indexed set. One of the most popular is Locality-sensitive hashing (LSH) by Charikar (2002), which is a binarization technique based on random projections that approximates the cosine similarity between two vectors through a monotonous function of the Hamming distance between the two corresponding binary codes. In our paper, LSH refers to this binarization strategy[2]. Many subsequent works have improved this initial binarization technique, such as spectal hashing (Weiss et al., 2009), or Iterative Quantization (ITQ) (Gong & Lazebnik, 2011), which learns a rotation matrix minimizing the quantization loss of the binarization. We refer the reader to two recent surveys by Wang et al. (2014) and Wang et al. (2015) for an overview of the binary hashing literature.

Beyond these binarization strategies, more general quantization techniques derived from Jegou et al. (2011) offer better trade-offs between memory and the approximation of a distance estimator. The Product Quantization (PQ) method approximates the distances by calculating, in the compressed domain, the distance between their quantized approximations. This method is statistically guaranteed to preserve the Euclidean distance between the vectors within an error bound directly related to the quantization error. The original PQ has been concurrently improved by Ge et al. (2013) and Norouzi & Fleet (2013), who learn an orthogonal transform minimizing the overall quantization loss. In our paper, we will consider the Optimized Product Quantization (OPQ) variant (Ge et al., 2013).

**Softmax approximation** The aforementioned works approximate either the Euclidean distance or the cosine similarity (both being equivalent in the case of unit-norm vectors). However, in the context of `fastText`, we are specifically interested in approximating the maximum inner product involved in a softmax layer. Several approaches derived from LSH have been recently proposed to achieve this goal, such as Asymmetric LSH by Shrivastava & Li (2014), subsequently discussed by Neyshabur & Srebro (2015). In our work, since we are not constrained to purely binary codes, we resort a more traditional encoding by employing a magnitude/direction parametrization of our vectors. Therefore we only need to encode/compress an unitary d-dimensional vector, which fits the aforementioned LSH and PQ methods well.

**Neural network compression models.** Recently, several research efforts have been conducted to compress the parameters of architectures involved in computer vision, namely for state-of-the-art Convolutional Neural Networks (CNNs) (Han et al., 2016; Lin et al., 2015). Some use vector quantization (Gong et al., 2014) while others binarize the network (Courbariaux et al., 2016). Denil et al. (2013) show that such classification models are easily compressed because they are over-parametrized, which concurs with early observations by LeCun et al. (1990).

---

[2]In the literature, LSH refers to multiple distinct strategies related to the Johnson-Lindenstrauss lemma. For instance, LSH sometimes refers to a partitioning technique with random projections allowing for sublinear search *via* cell probes, see for instance the $E^2$LSH variant of Datar et al. (2004).

Some of these works both aim at reducing the model size and the speed. In our case, since the `fastText` classifier on which our proposal is built upon is already very efficient, we are primilarly interested in reducing the size of the model while keeping a comparable classification efficiency.

## 3 PROPOSED APPROACH

### 3.1 TEXT CLASSIFICATION

In the context of text classification, linear classifiers (Joulin et al., 2016) remain competitive with more sophisticated, deeper models, and are much faster to train. On top of standard tricks commonly used in linear text classification (Agarwal et al., 2014; Wang & Manning, 2012; Weinberger et al., 2009), Joulin et al. (2016) use a low rank constraint to reduce the computation burden while sharing information between different classes. This is especially useful in the case of a large output space, where rare classes may have only a few training examples. In this paper, we focus on a similar model, that is, which minimizes the softmax loss $\ell$ over $N$ documents:

$$\sum_{n=1}^{N} \ell(y_n, BAx_n),\tag{1}$$

where $x_n$ is a bag of one-hot vectors and $y_n$ the label of the $n$-th document. In the case of a large vocabulary and a large output space, the matrices $A$ and $B$ are big and can require gigabytes of memory. Below, we describe how we reduce this memory usage.

### 3.2 BOTTOM-UP PRODUCT QUANTIZATION

**Product quantization** is a popular method for compressed-domain approximate nearest neighbor search (Jegou et al., 2011). As a compression technique, it approximates a real-valued vector by finding the closest vector in a pre-defined structured set of centroids, referred to as a codebook. This codebook is not enumerated, since it is extremely large. Instead it is implicitly defined by its structure: a $d$-dimensional vector $x \in \mathbb{R}^d$ is approximated as

$$\hat{x} = \sum_{i=1}^{k} q_i(x),\tag{2}$$

where the different subquantizers $q_i : x \mapsto q_i(x)$ are complementary in the sense that their respective centroids lie in distinct orthogonal subspaces, *i.e.*, $\forall i \neq j, \ \forall x, y, \ \langle q_i(x)|q_j(y)\rangle = 0$. In the original PQ, the subspaces are aligned with the natural axis, while OPQ learns a rotation, which amounts to alleviating this constraint and to not depend on the original coordinate system. Another way to see this is to consider that PQ splits a given vector $x$ into $k$ subvectors $x^i, i = 1 \ldots k$, each of dimension $d/k$: $x = [x^1 \ldots x^i \ldots x^k]$, and quantizes each sub-vector using a distinct k-means quantizer. Each subvector $x^i$ is thus mapped to the closest centroid amongst $2^b$ centroids, where $b$ is the number of bits required to store the quantization index of the subquantizer, typically $b = 8$. The reconstructed vector can take $2^{kb}$ distinct reproduction values, and is stored in $kb$ bits.

PQ estimates the inner product in the compressed domain as

$$x^\top y \approx \hat{x}^\top y = \sum_{i=1}^{k} q_i(x^i)^\top y^i.\tag{3}$$

This is a straightforward extension of the square L2 distance estimation of Jegou et al. (2011). In practice, the vector estimate $\hat{x}$ is trivially reconstructed from the codes, *i.e.*, from the quantization indexes, by concatenating these centroids.

The two parameters involved in PQ, namely the number of subquantizers $k$ and the number of bits $b$ per quantization index, are typically set to $k \in [2, d/2]$, and $b = 8$ to ensure byte-alignment.

*Discussion.* PQ offers several interesting properties in our context of text classification. Firstly, the training is very fast because the subquantizers have a small number of centroids, *i.e.*, 256 centroids for $b = 8$. Secondly, at test time it allows the reconstruction of the vectors with almost no

computational and memory overhead. Thirdly, it has been successfully applied in computer vision, offering much better performance than binary codes, which makes it a natural candidate to compress relatively shallow models. As observed by Sánchez & Perronnin (2011), using PQ just before the last layer incurs a very limited loss in accuracy when combined with a support vector machine.

In the context of text classification, the norms of the vectors are widely spread, typically with a ratio of 1000 between the max and the min. Therefore kmeans performs poorly because it optimizes an absolute error objective, so it maps all low-norm vectors to 0. A simple solution is to separate the norm and the angle of the vectors and to quantize them separately. This allows a quantization with no loss of performance, yet requires an extra $b$ bits per vector.

**Bottom-up strategy: re-training.** The first works aiming at compressing CNN models like the one proposed by (Gong et al., 2014) used the reconstruction from off-the-shelf PQ, *i.e.*, without any re-training. However, as observed in Sablayrolles et al. (2016), when using quantization methods like PQ, it is better to re-train the layers occurring after the quantization, so that the network can re-adjust itself to the quantization. There is a strong argument arguing for this re-training strategy: the square magnitude of vectors is reduced, on average, by the average quantization error for any quantizer satisfying the Lloyd conditions; see Jegou et al. (2011) for details.

This suggests a bottom-up learning strategy where we first quantize the input matrix, then retrain and quantize the output matrix (the input matrix being frozen). Experiments in section 4 show that it is worth adopting this strategy.

**Memory savings with PQ.** In practice, the bottom-up PQ strategy offers a compression factor of 10 without any noticeable loss of performance. Without re-training, we notice a drop in accuracy between $0.1\%$ and $0.5\%$, depending on the dataset and setting; see Section 4 and the appendix.

### 3.3 FURTHER TEXT SPECIFIC TRICKS

The memory usage strongly depends on the size of the vocabulary, which can be large in many text classification tasks. While it is clear that a large part of the vocabulary is useless or redundant, directly reducing the vocabulary to the most frequent words is not satisfactory: most of the frequent words, like "the" or "is" are not discriminative, in contrast to some rare words, *e.g.*, in the context of tag prediction. In this section, we discuss a few heuristics to reduce the space taken by the dictionary. They lead to major memory reduction, in extreme cases by a factor $100$. We experimentally show that this drastic reduction is complementary with the PQ compression method, meaning that the combination of both strategies reduces the model size by a factor up to $\times 1000$ for some datasets.

**Pruning the vocabulary.** Discovering which word or n-gram must be kept to preserve the overall performance is a feature selection problem. While many approaches have been proposed to select groups of variables during training (Bach et al., 2012; Meier et al., 2008), we are interested in selecting a fixed subset of $K$ words and ngrams from a pre-trained model. This can be achieved by selecting the $K$ embeddings that preserve as much of the model as possible, which can be reduced to selecting the $K$ words and ngrams associated with the highest norms.

While this approach offers major memory savings, it has one drawback occurring in some particular cases: some documents may not contained any of the $K$ best features, leading to a significant drop in performance. It is thus important to keep the $K$ best features under the condition that they cover the whole training set. More formally, the problem is to find a subset $\mathcal{S}$ in the feature set $\mathcal{V}$ that maximizes the sum of their norms $w_s$ under the constraint that all the documents in the training set $\mathcal{D}$ are covered:

$$\max_{\mathcal{S} \subseteq \mathcal{V}} \sum_{s \in \mathcal{S}} w_s \quad \text{s.t.} \quad |\mathcal{S}| \leq K, \quad P1_{\mathcal{S}} \geq 1_{\mathcal{D}},$$

where $P$ is a matrix such that $P_{ds} = 1$ if the $s$-th feature is in the $d$-th document, and 0 otherwise. This problem is directly related to set covering problems that are NP-hard (Feige, 1998). Standard greedy approaches require the storing of an inverted index or to do multiple passes over the dataset, which is prohibitive on very large dataset (Chierichetti et al., 2010). This problem can be cast as an instance of online submodular maximization with a rank constraint (Badanidiyuru et al., 2014;

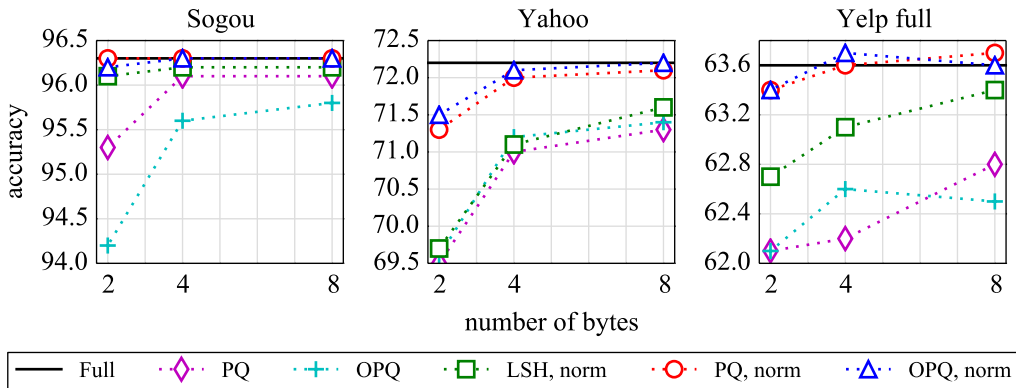

Figure 1: Accuracy as a function of the memory per vector/embedding on 3 datasets from Zhang et al. (2015). Note, an extra byte is required when we encode the norm explicitly ("norm").

Bateni et al., 2010). In our case, we use a simple online parallelizable greedy approach: For each document, we verify if it is already covered by a retained feature and, if not, we add the feature with the highest norm to our set of retained features. If the number of features is below $k$, we add the features with the highest norm that have not yet been picked.

**Hashing trick & Bloom filter.** On small models, the dictionary can take a significant portion of the memory. Instead of saving it, we extend the hashing trick used in Joulin et al. (2016) to both words and n-grams. This strategy is also used in Vowpal Wabbit (Agarwal et al., 2014) in the context of online training. This allows us to save around 1-2Mb with almost no overhead at test time (just the cost of computing the hashing function).

Pruning the vocabulary while using the hashing trick requires keeping a list of the indices of the $K$ remaining buckets. At test time, a binary search over the list of indices is required. It has a complexity of $O(\log(K))$ and a memory overhead of a few hundreds of kilobytes. Using Bloom filters instead reduces the complexity $\mathcal{O}(1)$ at test time and saves a few hundred kilobytes. However, in practice, it degrades performance.

## 4 EXPERIMENTS

This section evaluates the quality of our model compression pipeline and compare it to other compression methods on different text classification problems, and to other compact text classifiers.

**Evaluation protocol and datasets.** Our experimental pipeline is as follows: we train a model using `fastText` with the default setting unless specified otherwise. That is 2M buckets, a learning rate of 0.1 and 10 training epochs. The dimensionality $d$ of the embeddings is set to powers of 2 to avoid border effects that could make the interpretation of the results more difficult. As baselines, we use Locality-Sensitive Hashing (LSH) (Charikar, 2002), PQ (Jegou et al., 2011) and OPQ (Ge et al., 2013) (the non-parametric variant). Note that we use an improved version of LSH where random orthogonal matrices are used instead of random matrix projection Jégou et al. (2008). In a first series of experiments, we use the 8 datasets and evaluation protocol of Zhang et al. (2015). These datasets contain few million documents and have at most 10 classes. We also explore the limit of quantization on a dataset with an extremely large output space, that is a tag dataset extracted from the YFCC100M collection (Thomee et al., 2016)[3], referred to as FlickrTag in the rest of this paper.

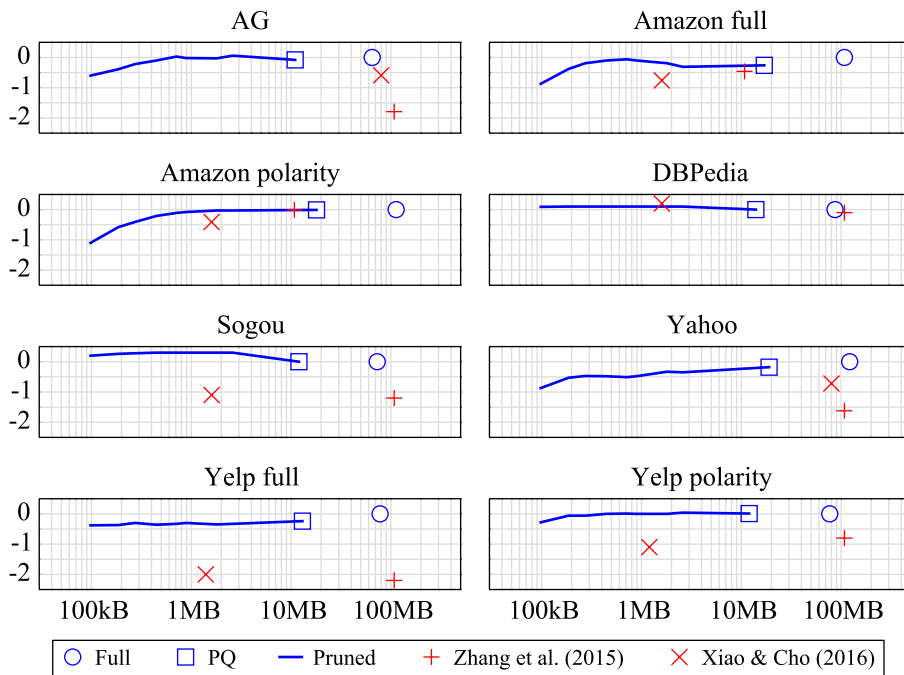

Figure 2: Loss of accuracy as a function of the model size. We compare the compressed model with different level of pruning with NPQ and the full `fastText` model. We also compare with Zhang et al. (2015) and Xiao & Cho (2016). Note that the size is in log scale.

## 4.1 SMALL DATASETS

**Compression techniques.** We compare three popular methods used for similarity estimation with compact codes: LSH, PQ and OPQ on the datasets released by Zhang et al. (2015). Figure 1 shows the accuracy as a function of the number of bytes used per embedding, which corresponds to the number $k$ of subvectors in the case of PQ and OPQ. See more results in the appendix. As discussed in Section 2, LSH reproduces the cosine similarity and is therefore not adapted to un-normalized data. Therefore we only report results with normalization. Once normalized, PQ and OPQ are almost lossless even when using only $k = 4$ subquantizers per embedding (equivalently, bytes). We observe in practice that using $k = d/2$, *i.e.*, half of the components of the embeddings, works well in practice. In the rest of the paper and if not stated otherwise, we focus on this setting. The difference between the normalized versions of PQ and OPQ is limited and depends on the dataset. Therefore we adopt the normalized PQ (NPQ) for the rest of this study, since it is faster to train.

| word | Entropy | Norm | word | Entropy | Norm |
|------|---------|------|------|---------|------|
| . | 1 | 354 | mediocre | 1399 | 1 |
| , | 2 | 176 | disappointing | 454 | 2 |
| the | 3 | 179 | so-so | 2809 | 3 |
| and | 4 | 1639 | lacks | 1244 | 4 |
| i | 5 | 2374 | worthless | 1757 | 5 |
| a | 6 | 970 | dreadful | 4358 | 6 |
| to | 7 | 1775 | drm | 6395 | 7 |
| it | 8 | 1956 | poorly | 716 | 8 |
| of | 9 | 2815 | uninspired | 4245 | 9 |
| this | 10 | 3275 | worst | 402 | 10 |

Table 1: Best ranked words w.r.t. entropy (*left*) and norm (*right*) on the Amazon full review dataset. We give the rank for both criteria. The norm ranking filters out words carrying little information.

---

[3]Data available at https://research.facebook.com/research/fasttext/

| Dataset | | full | 64KiB | 32KiB | 16 KiB |
|---|---|---|---|---|---|
| AG | 65M | 92.1 | 91.4 | 90.6 | 89.1 |
| Amazon full | 108M | 60.0 | 58.8 | 56.0 | 52.9 |
| Amazon pol. | 113M | 94.5 | 93.3 | 92.1 | 89.3 |
| DBPedia | 87M | 98.4 | 98.2 | 98.1 | 97.4 |
| Sogou | 73M | 96.4 | 96.4 | 96.3 | 95.5 |
| Yahoo | 122M | 72.1 | 70.0 | 69.0 | 69.2 |
| Yelp full | 78M | 63.8 | 63.2 | 62.4 | 58.7 |
| Yelp pol. | 77M | 95.7 | 95.3 | 94.9 | 93.2 |
| Average diff. [%] | | 0 | -0.8 | -1.7 | -3.5 |

Table 2: Performance on very small models. We use a quantization with $k = 1$, hashing and an extreme pruning. The last row shows the average drop of performance for different size.

**Pruning.** Figure 2 shows the performance of our model with different sizes. We fix $k = d/2$ and use different pruning thresholds. NPQ offers a compression rate of $\times 10$ compared to the full model. As the pruning becomes more agressive, the overall compression can increase up to $\times 1,000$ with little drop of performance and no additional overhead at test time. In fact, using a smaller dictionary makes the model *faster* at test time. We also compare with character-level Convolutional Neural Networks (CNN) (Zhang et al., 2015; Xiao & Cho, 2016). They are attractive models for text classification because they achieve similar performance with less memory usage than linear models (Xiao & Cho, 2016). Even though `fastText` with the default setting uses more memory, NPQ is already on par with CNNs' memory usage. Note that CNNs are not quantized, and it would be worth seeing how much they can be quantized with no drop of performance. Such a study is beyond the scope of this paper. Our pruning is based on the norm of the embeddings according to the guidelines of Section 3.3. Table 1 compares the ranking obtained with norms to the ranking obtained using entropy, which is commonly used in unsupervised settings Stolcke (2000).

**Extreme compression.** Finally, in Table 2, we explore the limit of quantized model by looking at the performance obtained for models under 64KiB. Surprisingly, even at 64KiB and 32KiB, the drop of performance is only around 0.8% and 1.7% despite a compression rate of $\times 1,000 - 4,000$.

## 4.2 LARGE DATASET: FLICKRTAG

In this section, we explore the limit of compression algorithms on very large datasets. Similar to Joulin et al. (2016), we consider a hashtag prediction dataset containing $312,116$ labels. We set the minimum count for words at 10, leading to a dictionary of $1,427,667$ words. We take 10M buckets for n-grams and a hierarchical softmax. We refer to this dataset as FlickrTag.

**Output encoding.** We are interested in understanding how the performance degrades if the classifier is also quantized (*i.e.*, the matrix $B$ in Eq. 1) and when the pruning is at the limit of the minimum number of features required to cover the full dataset.

| Model | $k$ | norm | retrain | Acc. | Size |
|---|---|---|---|---|---|
| full (uncompressed) | | | | 45.4 | 12 GiB |
| Input | 128 | | | 45.0 | 1.7 GiB |
| Input | 128 | x | | 45.3 | 1.8 GiB |
| Input | 128 | x | x | 45.5 | 1.8 GiB |
| Input+Output | 128 | x | | 45.2 | 1.5 GiB |
| Input+Output | 128 | x | x | 45.4 | 1.5 GiB |

Table 3: FlickrTag: Influence of quantizing the output matrix on performance. We use PQ for quantization with an optional normalization. We also retrain the output matrix after quantizing the input one. The "norm" refers to the separate encoding of the magnitude and angle, while "retrain" refers to the re-training bottom-up PQ method described in Section 3.2.

Table 3 shows that quantizing both the "input" matrix (*i.e.*, $A$ in Eq. 1) and the "output" matrix (*i.e.*, $B$) does not degrade the performance compared to the full model. We use embeddings with $d = 256$ dimensions and use $k = d/2$ subquantizers. We do not use any text specific tricks, which leads to a compression factor of $8$. Note that even if the output matrix is not retrained over the embeddings, the performance is only $0.2\%$ away from the full model. As shown in the Appendix, using less subquantizers significantly decreases the performance for a small memory gain.

| Model | full | Entropy pruning | | Norm pruning | | Max-Cover pruning | |
|---|---|---|---|---|---|---|---|
| #embeddings | 11.5M | 2M | 1M | 2M | 1M | 2M | 1M |
| Memory | 12GiB | 297MiB | 174MiB | 305MiB | 179MiB | 305MiB | 179MiB |
| Coverage [%] | 88.4 | 70.5 | 70.5 | 73.2 | 61.9 | 88.4 | 88.4 |
| Accuracy | 45.4 | 32.1 | 30.5 | 41.6 | 35.8 | 45.5 | 43.9 |

Table 4: FlickrTag: Comparison of entropy pruning, norm pruning and max-cover pruning methods. We show the coverage of the test set for each method.

**Pruning.**   Table 4 shows how the performance evolves with pruning. We measure this effect on top of a fully quantized model. The full model misses $11.6\%$ of the test set because of missing words (some documents are either only composed of hashtags or have only rare words). There are $312,116$ labels and thus it seems reasonable to keep embeddings in the order of the million. A naive pruning with 1M features misses about $30-40\%$ of the test set, leading to a significant drop of performance. On the other hand, even though the max-coverage pruning approach was set on the train set, it does not suffer from any coverage loss on the test set. This leads to a smaller drop of performance. If the pruning is too aggressive, however, the coverage decreases significantly.

## 5   FUTURE WORK

It may be possible to obtain further reduction of the model size in the future. One idea is to condition the size of the vectors (both for the input features and the labels) based on their frequency (Chen et al., 2015; Grave et al., 2016). For example, it is probably not worth representing the rare labels by full 256-dimensional vectors in the case of the FlickrTag dataset. Thus, conditioning the vector size on the frequency and norm seems like an interesting direction to explore in the future.

We may also consider combining the entropy and norm pruning criteria: instead of keeping the features in the model based just on the frequency or the norm, we can use both to keep a good set of features. This could help to keep features that are both frequent and discriminative, and thereby to reduce the coverage problem that we have observed.

Additionally, instead of pruning out the less useful features, we can decompose them into smaller units (Mikolov et al., 2012). For example, this can be achieved by splitting every non-discriminative word into a sequence of character trigrams. This could help in cases where training and test examples are very short (for example just a single word).

## 6   CONCLUSION

In this paper, we have presented several simple techniques to reduce, by several orders of magnitude, the memory complexity of certain text classifiers without sacrificing accuracy nor speed. This is achieved by applying discriminative pruning which aims to keep only important features in the trained model, and by performing quantization of the weight matrices and hashing of the dictionary.

We will publish the code as an extension of the `fastText` library. We hope that our work will serve as a baseline to the research community, where there is an increasing interest for comparing the performance of various deep learning text classifiers for a given number of parameters. Overall, compared to recent work based on convolutional neural networks, `fastText.zip` is often more accurate, while requiring several orders of magnitude less time to train on common CPUs, and incurring a fraction of the memory complexity.

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

APPENDIX

In the appendix, we show some additional results. The model used in these experiments only had 1M ngram buckets. In Table 5, we show a thorough comparison of LSH, PQ and OPQ on 8 different datasets. Table 7 summarizes the comparison with CNNs in terms of accuracy and size. Table 8 show a thorough comparison of the hashing trick and the Bloom filters.

| Quant. | k | norm | AG | | Amz. f. | | Amz. p. | | DBP | | Sogou | | Yah. | | Yelp f. | | Yelp p. | |
|---|---|---|---|---|---|---|---|---|---|---|---|---|---|---|---|---|---|---|
| full | | | 92.1 | 36M | 59.8 | 97M | 94.5 | 104M | 98.4 | 67M | 96.3 | 47M | 72 | 120M | 63.7 | 56M | 95.7 | 53M |
| full,nodict | | | 92.1 | 34M | 59.9 | 78M | 94.5 | 83M | 98.4 | 56M | 96.3 | 42M | 72.2 | 91M | 63.6 | 48M | 95.6 | 46M |
| LSH | 8 | | 88.7 | 8.5M | 51.3 | 20M | 90.3 | 21M | 92.7 | 14M | 94.2 | 11M | 54.8 | 23M | 56.7 | 12M | 92.2 | 12M |
| PQ | 8 | | 91.7 | 8.5M | 59.3 | 20M | 94.4 | 21M | 97.4 | 14M | 96.1 | 11M | 71.3 | 23M | 62.8 | 12M | 95.4 | 12M |
| OPQ | 8 | | 91.9 | 8.5M | 59.3 | 20M | 94.4 | 21M | 96.9 | 14M | 95.8 | 11M | 71.4 | 23M | 62.5 | 12M | 95.4 | 12M |
| LSH | 8 | x | 91.9 | 9.5M | 59.4 | 22M | 94.5 | 24M | 97.8 | 16M | 96.2 | 12M | 71.6 | 26M | 63.4 | 14M | 95.6 | 13M |
| PQ | 8 | x | 92.0 | 9.5M | 59.8 | 22M | 94.5 | 24M | 98.4 | 16M | 96.3 | 12M | 72.1 | 26M | 63.7 | 14M | 95.6 | 13M |
| OPQ | 8 | x | 92.1 | 9.5M | 59.9 | 22M | 94.5 | 24M | 98.4 | 16M | 96.3 | 12M | 72.2 | 26M | 63.6 | 14M | 95.6 | 13M |
| LSH | 4 | | 88.3 | 4.3M | 50.5 | 9.7M | 88.9 | 11M | 91.6 | 7.0M | 94.3 | 5.3M | 54.6 | 12M | 56.5 | 6.0M | 92.9 | 5.7M |
| PQ | 4 | | 91.6 | 4.3M | 59.2 | 9.7M | 94.4 | 11M | 96.3 | 7.0M | 96.1 | 5.3M | 71.0 | 12M | 62.2 | 6.0M | 95.4 | 5.7M |
| OPQ | 4 | | 91.7 | 4.3M | 59.0 | 9.7M | 94.4 | 11M | 96.9 | 7.0M | 95.6 | 5.3M | 71.2 | 12M | 62.6 | 6.0M | 95.4 | 5.7M |
| LSH | 4 | x | 92.1 | 5.3M | 59.2 | 13M | 94.4 | 13M | 97.7 | 8.8M | 96.2 | 6.6M | 71.1 | 15M | 63.1 | 7.4M | 95.5 | 7.2M |
| PQ | 4 | x | 92.1 | 5.3M | 59.8 | 13M | 94.5 | 13M | 98.4 | 8.8M | 96.3 | 6.6M | 72.0 | 15M | 63.6 | 7.5M | 95.6 | 7.2M |
| OPQ | 4 | x | 92.2 | 5.3M | 59.8 | 13M | 94.5 | 13M | 98.3 | 8.8M | 96.3 | 6.6M | 72.1 | 15M | 63.7 | 7.5M | 95.6 | 7.2M |
| LSH | 2 | | 87.7 | 2.2M | 50.1 | 4.9M | 88.9 | 5.2M | 90.6 | 3.5M | 93.9 | 2.7M | 51.4 | 5.7M | 56.6 | 3.0M | 91.3 | 2.9M |
| PQ | 2 | | 91.1 | 2.2M | 58.7 | 4.9M | 94.4 | 5.2M | 87.1 | 3.6M | 95.3 | 2.7M | 69.5 | 5.7M | 62.1 | 3.0M | 95.4 | 2.9M |
| OPQ | 2 | | 91.4 | 2.2M | 58.2 | 4.9M | 94.3 | 5.2M | 91.6 | 3.6M | 94.2 | 2.7M | 69.6 | 5.7M | 62.1 | 3.0M | 95.4 | 2.9M |
| LSH | 2 | x | 91.8 | 3.2M | 58.6 | 7.3M | 94.3 | 7.8M | 97.1 | 5.3M | 96.1 | 4.0M | 69.7 | 8.6M | 62.7 | 4.5M | 95.5 | 4.3M |
| PQ | 2 | x | 91.9 | 3.2M | 59.6 | 7.3M | 94.5 | 7.8M | 98.1 | 5.3M | 96.3 | 4.0M | 71.3 | 8.6M | 63.4 | 4.5M | 95.6 | 4.3M |
| OPQ | 2 | x | 92.1 | 3.2M | 59.5 | 7.3M | 94.5 | 7.8M | 98.1 | 5.3M | 96.2 | 4.0M | 71.5 | 8.6M | 63.4 | 4.5M | 95.6 | 4.3M |

Table 5: Comparison between standard quantization methods. The original model has a dimensionality of 8 and 2M buckets. Note that all of the methods are without dictionary.

| k | co | AG | | Amz. f. | | Amz. p. | | DBP | | Sogou | | Yah. | | Yelp f. | | Yelp p. | |
|---|---|---|---|---|---|---|---|---|---|---|---|---|---|---|---|---|---|
| full, nodict | | 92.1 | 34M | 59.8 | 78M | 94.5 | 83M | 98.4 | 56M | 96.3 | 42M | 72.2 | 91M | 63.7 | 48M | 95.6 | 46M |
| 8 | full | 92.0 | 9.5M | 59.8 | 22M | 94.5 | 24M | 98.4 | 16M | 96.3 | 12M | 72.1 | 26M | 63.7 | 14M | 95.6 | 13M |
| 4 | full | 92.1 | 5.3M | 59.8 | 13M | 94.5 | 13M | 98.4 | 8.8M | 96.3 | 6.6M | 72 | 15M | 63.6 | 7.5M | 95.6 | 7.2M |
| 2 | full | 91.9 | 3.2M | 59.6 | 7.3M | 94.5 | 7.8M | 98.1 | 5.3M | 96.3 | 4.0M | 71.3 | 8.6M | 63.4 | 4.5M | 95.6 | 4.3M |
| 8 | 200K | 92.0 | 2.5M | 59.7 | 2.5M | 94.3 | 2.5M | 98.5 | 2.5M | 96.6 | 2.5M | 71.8 | 2.5M | 63.3 | 2.5M | 95.6 | 2.5M |
| 8 | 100K | 91.9 | 1.3M | 59.5 | 1.3M | 94.3 | 1.3M | 98.5 | 1.3M | 96.6 | 1.3M | 71.6 | 1.3M | 63.4 | 1.3M | 95.6 | 1.3M |
| 8 | 50K | 91.7 | 645K | 59.7 | 645K | 94.3 | 644K | 98.5 | 645K | 96.6 | 645K | 71.5 | 645K | 63.2 | 645K | 95.6 | 644K |
| 8 | 10K | 91.3 | 137K | 58.6 | 137K | 93.2 | 137K | 98.5 | 137K | 96.5 | 137K | 71.3 | 137K | 63.3 | 137K | 95.4 | 137K |
| 4 | 200K | 92.0 | 1.8M | 59.7 | 1.8M | 94.3 | 1.8M | 98.5 | 1.8M | 96.6 | 1.8M | 71.7 | 1.8M | 63.3 | 1.8M | 95.6 | 1.8M |
| 4 | 100K | 91.9 | 889K | 59.5 | 889K | 94.4 | 889K | 98.5 | 889K | 96.6 | 889K | 71.7 | 889K | 63.4 | 889K | 95.6 | 889K |
| 4 | 50K | 91.7 | 449K | 59.6 | 449K | 94.3 | 449K | 98.5 | 450K | 96.6 | 449K | 71.4 | 450K | 63.2 | 449K | 95.5 | 449K |
| 4 | 10K | 91.5 | 98K | 58.6 | 98K | 93.2 | 98K | 98.5 | 98K | 96.5 | 98K | 71.2 | 98K | 63.3 | 98K | 95.4 | 98K |
| 2 | 200K | 91.9 | 1.4M | 59.6 | 1.4M | 94.3 | 1.4M | 98.4 | 1.4M | 96.5 | 1.4M | 71.5 | 1.4M | 63.2 | 1.4M | 95.5 | 1.4M |
| 2 | 100K | 91.6 | 693K | 59.5 | 693K | 94.3 | 693K | 98.4 | 694K | 96.6 | 693K | 71.1 | 694K | 63.2 | 693K | 95.6 | 693K |
| 2 | 50K | 91.6 | 352K | 59.6 | 352K | 94.3 | 352K | 98.4 | 352K | 96.5 | 352K | 71.1 | 352K | 63.2 | 352K | 95.6 | 352K |
| 2 | 10K | 91.3 | 78K | 58.5 | 78K | 93.2 | 78K | 98.4 | 79K | 96.5 | 78K | 70.8 | 78K | 63.2 | 78K | 95.3 | 78K |

Table 6: Comparison with different quantization and level of pruning. "co" is the cut-off parameter of the pruning.

| Dataset | Zhang et al. (2015) | | Xiao & Cho (2016) | | `fastText`+PQ, $k = d/2$ | |
|---------|-----|------|------|-----|------|------|
| AG | 90.2 | 108M | 91.4 | 80M | 91.9 | 889K |
| Amz. f. | 59.5 | 10.8M | 59.2 | 1.6M | 59.6 | 449K |
| Amz. p. | 94.5 | 10.8M | 94.1 | 1.6M | 94.3 | 449K |
| DBP | 98.3 | 108M | 98.6 | 1.2M | 98.5 | 98K |
| Sogou | 95.1 | 108M | 95.2 | 1.6M | 96.5 | 98K |
| Yah. | 70.5 | 108M | 71.4 | 80M | 71.7 | 889K |
| Yelp f. | 61.6 | 108M | 61.8 | 1.4M | 63.3 | 98K |
| Yelp p. | 94.8 | 108M | 94.5 | 1.2M | 95.5 | 449K |

Table 7: Comparison between CNNs and `fastText` with and without quantization. The numbers for Zhang et al. (2015) are reported from Xiao & Cho (2016). Note that for the CNNs, we report the size of the model under the assumption that they use float32 storage. For `fastText`(+PQ) we report the memory used in RAM at test time.

| Quant. | Bloom | co | AG | | Amz. f. | | Amz. p. | | DBP | | Sogou | | Yah. | | Yelp f. | | Yelp p. | |
|--------|-------|------|------|-----|------|-----|------|-----|------|-----|------|-----|------|-----|------|-----|------|-----|
| full,nodict | | | 92.1 | 34M | 59.8 | 78M | 94.5 | 83M | 98.4 | 56M | 96.3 | 42M | 72.2 | 91M | 63.7 | 48M | 95.6 | 46M |
| NPQ | | 200K | 91.9 | 1.4M | 59.6 | 1.4M | 94.3 | 1.4M | 98.4 | 1.4M | 96.5 | 1.4M | 71.5 | 1.4M | 63.2 | 1.4M | 95.5 | 1.4M |
| NPQ | x | 200K | 92.2 | 830K | 59.3 | 830K | 94.1 | 830K | 98.4 | 830K | 96.5 | 830K | 70.7 | 830K | 63.0 | 830K | 95.5 | 830K |
| NPQ | | 100K | 91.6 | 693K | 59.5 | 693K | 94.3 | 693K | 98.4 | 694K | 96.6 | 693K | 71.1 | 694K | 63.2 | 693K | 95.6 | 693K |
| NPQ | x | 100K | 91.8 | 420K | 59.1 | 420K | 93.9 | 420K | 98.4 | 420K | 96.5 | 420K | 70.6 | 420K | 62.8 | 420K | 95.3 | 420K |
| NPQ | | 50K | 91.6 | 352K | 59.6 | 352K | 94.3 | 352K | 98.4 | 352K | 96.5 | 352K | 71.1 | 352K | 63.2 | 352K | 95.6 | 352K |
| NPQ | x | 50K | 91.5 | 215K | 58.8 | 215K | 93.6 | 215K | 98.3 | 215K | 96.5 | 215K | 70.1 | 215K | 62.7 | 215K | 95.1 | 215K |
| NPQ | | 10K | 91.3 | 78K | 58.5 | 78K | 93.2 | 78K | 98.4 | 79K | 96.5 | 78K | 70.8 | 78K | 63.2 | 78K | 95.3 | 78K |
| NPQ | x | 10K | 90.8 | 51K | 56.8 | 51K | 91.7 | 51K | 98.1 | 51K | 96.1 | 51K | 68.7 | 51K | 61.7 | 51K | 94.5 | 51K |

Table 8: Comparison with and without Bloom filters. For NPQ, we set $d = 8$ and $k = 2$.

| Model | k | norm | retrain | Acc. | Size |
|---|---|---|---|---|---|
| full | | | | 45.4 | 12G |
| Input | 128 | | | 45.0 | 1.7G |
| Input | 128 | x | | 45.3 | 1.8G |
| Input | 128 | x | x | 45.5 | 1.8G |
| Input+Output | 128 | x | | 45.2 | 1.5G |
| Input+Output | 128 | x | x | 45.4 | 1.5G |
| Input+Output, co=2M | 128 | x | x | 45.5 | 305M |
| Input+Output, n co=1M | 128 | x | x | 43.9 | 179M |
| Input | 64 | | | 44.0 | 1.1G |
| Input | 64 | x | | 44.7 | 1.1G |
| Input | 64 | x | | 44.9 | 1.1G |
| Input+Output | 64 | x | | 44.6 | 784M |
| Input+Output | 64 | x | x | 44.8 | 784M |
| Input+Output, co=2M | 64 | x | | 42.5 | 183M |
| Input+Output, co=1M | 64 | x | | 39.9 | 118M |
| Input+Output, co=2M | 64 | x | x | 45.0 | 183M |
| Input+Output, co=1M | 64 | x | x | 43.4 | 118M |
| Input | 32 | | | 40.5 | 690M |
| Input | 32 | x | | 42.4 | 701M |
| Input | 32 | x | x | 42.9 | 701M |
| Input+Output | 32 | x | | 42.3 | 435M |
| Input+Output | 32 | x | x | 42.8 | 435M |
| Input+Output, co=2M | 32 | x | | 35.0 | 122M |
| Input+Output, co=1M | 32 | x | | 32.6 | 88M |
| Input+Output, co=2M | 32 | x | x | 43.3 | 122M |
| Input+Output, co=1M | 32 | x | x | 41.6 | 88M |

Table 9: FlickrTag: Comparison for a large dataset of (i) different quantization methods and parameters, (ii) with or without re-training.

