# Peer review of "FastText.zip: Compressing text classification models"

_ICLR 2017 — rejected_

[Official Review · AnonReviewer2 · rating 6 · confidence 3 · 15 Dec 2016]
**Lossy compression techniques applied to FastText with nice results**

This paper describes how to approximate the FastText approach such that its memory footprint is reduced by several orders of magnitude, while preserving its classification accuracy. The original FastText approach was based on a linear classifier on top of bag-of-words embeddings. This type of method is extremely fast to train and test, but the model size can be quite large.

This paper focuses on approximating the original approach with lossy compression techniques. Namely, the embeddings and classifier matrices A and B are compressed with Product Quantization, and an aggressive dictionary pruning is carried out. Experiments on various datasets (either with small or large number of classes) are conducted to tune the parameters and demonstrate the effectiveness of the approach. With a negligible loss in classification accuracy, an important reduction in term of model size (memory footprint) can be achieved, in the order of 100~1000 folds compared to the original size.

The paper is well written overall. The goal is clearly defined and well carried out, as well as the experiments. Different options for compressing the model data are evaluated and compared (e.g. PQ vs LSH), which is also interesting. Nevertheless the paper does not propose by itself any novel idea for text classification. It just focuses on adapting existing lossy compression techniques, which is not necessarily a problem. Specifically, it introduces:
  - a straightforward variant of PQ for unnormalized vectors,
  - dictionary pruning is cast as a set covering problem (which is NP-hard), but a greedy approach is shown to yield excellent results nonetheless,
  - hashing tricks and bloom filter are simply borrowed from previous papers.

These techniques are quite generic and could as well be used in other works. 


Here are some minor problems with the paper:

  - it is not made clear how the full model size is computed. What is exactly in the model? Which proportion of the full size do the A and B matrices, the dictionary, and the rest, account for? It is hard to follow where is the size bottleneck, which also seems to depend on the target application (i.e. small or large number of test classes). It would have been nice to provide a formula to calculate the total model size as a function of all parameters (k,b for PQ and K for dictionary, number of classes).
  
  - some parts lack clarity. For instance, the greedy approach to prune the dictionary is exposed in less than 4 lines (top of page 5), though it is far from being straightforward. Likewise, it is not clear why the binary search used for the hashing trick would introduce an overhead of a few hundreds of KB.
  

Overall this looks like a solid work, but with potentially limited impact research-wise.

[Official Review · AnonReviewer1 · rating 6 · confidence 4 · 19 Dec 2016]
**Effective if simple combination of existing techniques for text-classifier compression**

The paper proposes a series of tricks for compressing fast (linear) text classification models. The paper is clearly written, and the results are quite strong. The main compression is achieved via product quantization, a technique which has been explored in other applications within the neural network model compression literature. In addition to the Gong et al. work which was cited, it would be worth mentioning Quantized Convolutional Neural Networks for Mobile Devices (CVPR 2016,

[Official Review · AnonReviewer3 · rating 5 · confidence 4 · 23 Dec 2016]

The paper presents a few tricks to compress a wide and shallow text classification model based on n-gram features. These tricks include (1) using (optimized) product quantization to compress embedding weights (2) pruning some of the vocabulary elements (3) hashing to reduce the storage of the vocabulary (this is a minor component of the paper). The paper focuses on models with very large vocabularies and shows a reduction in the size of the models at a relatively minor reduction of the accuracy.

The problem of compressing neural models is important and interesting. The methods section of the paper is well written with good high level comments and references. However, the machine learning contributions of the paper are marginal to me. The experiments are not too convincing mainly focusing on benchmarks that are not commonly used. The implications of the paper on the state-of-the-art RNN text classification models is unclear.

The use of (optimized) product quantization for approximating inner product is not particularly novel. Previous work also considered doing this. Most of the reduction in the model sizes comes from pruning vocabulary elements. The method proposed for pruning vocabulary elements is simply based on the assumption that embeddings with larger L2 norm are more important. A coverage heuristic is taken into account too. From a machine learning point of view, the proper baseline to solve this problem is to have a set of (relaxed) binary coefficients for each embedding vector and learn the coefficients jointly with the weights. An L1 regularizer on the coefficients can be used to encourage sparsity. From a practical point of view, I believe an important baseline is missing: what if one simply uses fewer vocabulary elements (e.g based on subword units - see

[Final Decision · Program Chairs · 06 Feb 2017]
**ICLR committee final decision**

The submission describes a method for compressing shallow and wide text classification models. The paper is well written, but the proposed method is not particularly novel, as it's comprised of existing model compression techniques. Overall, the contributions are incremental and the potential impact seems rather limited.